**Data Availability Statement:** All relevant data are within the paper and its Supporting Information.

# Improving case-detection of severe wasting among under-five-year-old children in Timor Leste: A secondary analysis of data from the 2020 national cross-sectional food and nutrition survey

**Mueni Mutunga**[1], **Faraja Chiwile**[2], **Natalia dos Reis de Araujo Moniz**[3], **Paluku Bahwere**[4] *

1 United Nations Children's Fund (UNICEF) East Asia Pacific Regional Office, Bangkok, Thailand, 2 United Nations Children's Fund (UNICEF), Timor-Lest Country Office, Dili, Democratic Republic of Timor-Leste, 3 Ministry of Health, Democratic Republic of Timor-Leste, Dili, Democratic Republic of Timor-Leste, 4 Action Against Hunger UK, London, United Kingdom

☯ These authors contributed equally to this work.
* paluku.bahwere@gmail.com

## Abstract

The World Health Organization recommends using weight-for-height Z-score (WHZ) <-3 or Mid-Upper Arm Circumference (MUAC) <115 mm as independent criteria for diagnosing severe wasting. However, there are several challenges in using the WHZ criterion. As a result, the MUAC (and edema)-only approach for identifying children needing treatment for severe wasting has been developed and is being rapidly scaled-up globally, including in Timor-Leste. But previous studies reported that MUAC<115 mm has poor diagnostic accuracy for detecting children with WHZ<-3. The two options being explored globally for improving the identification of these children in MUAC (and edema)-only programming contexts include expanding MUAC cut-off and the combination of the indicators MUAC and Weight-for-Age Z-score (WAZ). This study explored the accuracy for diagnosing severe wasting (WHZ<-3) of these two options in Timor-Leste. We conducted a secondary analysis of data from the 2020 national Timor-Leste Food and Nutrition Survey. We tested the accuracy of various MUAC cut-offs, and predefined case definitions in five age groups (0–5 months, 6–23 months, 24–59 months, 6–59 months, and 0–59 months). We calculated the standard diagnostic test parameters (sensitivity, specificity, Youden Index, and others) and used the Youden Index as the principal criterion for rating the overall level of accuracy. The sample analyzed comprised 11,056 children with complete information on our key variables (anthropometric data, age, and sex), of whom 52.2% were boys. The age groups 0 to 5 months, 6 to 23 months, and 24 to 59 months represented 9.0%, 33.7%, and 57.3% of the sample, respectively. We found that the optimal diagnostic MUAC cut-off varied across the age groups between 117 mm and 142 mm, with the Youden Index remaining < 55% in all the age groups considered. The use of case definitions combing MUAC and WAZ optimized the identification of children with WHZ<-3. The case definition MUAC<130 mm or WAZ<-3

**Funding:** The survey that generated the dataset used for this study was funded by the government of the Democratic Republic of Timor-Leste and the European Union. The funders played no role in the design, the data analysis, the interpretation of the findings, and the manuscript preparation.

**Competing interests:** The authors have declared that not competing interests exist

Z-score had the best diagnostic accuracy in all the age groups except for the 0 to 5 months age group for which the case definition MUAC<110 mm or WAZ<-2 Z-score had the highest Youden Index. Our findings show that it is challenging to significantly improve diagnostic accuracy for identifying children with WHZ<-3 by only expanding the MUAC cut-off in under five Timorese children. However, In settings facing challenges in using WHZ, the combination of MUAC and WAZ indicators offers a promising approach. Further research is needed to confirm the effectiveness of the proposed combination of MUAC and WAZ indicators case definitions in a programmatic context in Timor-Leste, and other similar contexts.

## Background

Wasting is a condition that relates to a recent and severe weight loss [1–3]. Its immediate causes are an inadequate quality and quantity of a person's diet or frequent or prolonged illnesses [1–3]. It is a global public health issue that affects several million children under the age of five years worldwide [4]. By the latest 2022 estimates of the indicator weight-for-height Z-score, 45 million under five years children were wasted, of which 13.7 million were severely wasted [4]. This figure does not include the number of children suffering from edematous malnutrition and children severely wasted by MUAC, not WHZ which means that the global burden of severe acute malnutrition (SAM) is higher [5]. Wasting in children is associated with a nine-fold higher risk of death if not appropriately treated, but its case-fatality is considerably reduced if treated with the currently recommended protocols that comprise the prevention and treatment of associated infections and micronutrients disturbance and correcting weight deficit by using specially designed foods [6–10].

According to the World Health Organization (WHO) guidelines, severe wasting has historically been defined as low weight-for-height Z-score (WHZ<-3) [11]. Mid-upper arm circumference (MUAC) was later introduced as an additional independent anthropometric criterion with the cut-off of MUAC<115 mm proposed for the diagnosis of severe wasting [12–15]. In the past, classifications based on the Weight-for-Age Z-score (WAZ) indicator were also widely used but they are almost no longer used in acute malnutrition programming [12, 16]. This is despite that this indicator is still widely used in growth monitoring and promotion programs (GMP) which are among the rare nutrition interventions well-established and fully-integrated in national health systems in most countries in which under-five wasting is a public health concern [17, 18]. In accordance with global policies and guidelines, Timor-Leste national guidelines for wasting identification and management recommend using both WHZ and MUAC indicators but not WAZ indicator despite the countrywide implementation of the GMP. However, it has been difficult to consistently use the WHZ indicator, given the challenges of measuring length/height in decentralized health facilities. Thus, the Integrated Management of Acute Malnutrition (IMAM) program uses MUAC as the sole anthropometric indicator for identifying, enrolling, and discharging wasted children.

MUAC and WHZ thresholds are integrated in emergency nutrition assessments and national treatment protocols as independent criterion for the diagnosis of wasting among children aged between 6 to 59 months [10, 13, 15]. WHO has also recently recommended MUAC<110 mm as criterion for diagnosing risk of poor growth and development including severe wasting in infants below 6 months of age [10]. While wasting and acute malnutrition are often interchangeable terms, acute Malnutrition can also be diagnosed from clinical abnormalities, most notably the presence of bilateral pitting edema [13]. This paper will focus on the

utility of anthropometric indicators for wasting (WHZ and MUAC) and not on acute malnutrition or edema.

While it is often assumed to be a close relationship between WHZ and MUAC, the diagnostic criteria based on the two indicators classify different groups of children as wasted, with MUAC-based criterion misclassifying a substantial proportion, varying across contexts, of children diagnosed as wasted by the WHZ-based criterion and vice-versa [19–26].

The discordance also affects the estimation of the prevalence of wasting, with WHZ often giving a higher prevalence than MUAC, with the magnitude of the difference varying widely across contexts [27, 28]. WHO recommends the use in programs addressing wasting both WHZ and MUAC, but operational difficulties have been precluding the implementation of this recommendation. MUAC is a simple and inexpensive measure that is widely used to diagnose severe wasting in the community and that can be used by low literate trained community health workers (CHWs) and parents of malnourished children [29–32]. MUAC measurements done with a simple MUAC tape are quicker and easier to implement than WHZ that require scales for weight and boards for height measurements and interpretation of the two measures in comparison with expected values provided by the standard growth curve [33, 34]. In some contexts, the use of the WHZ indicator is not feasible because of operational constraints including shortage of skilled staffs, transportation of needed equipment, challenges and shortage of appropriate anthropometric equipment [14, 35, 36]. The challenges of using the WHZ indicator in some contexts contributed to the development and expansion of the MUAC(and edema)-only approach that uses MUAC as the sole anthropometric for diagnosing wasting and edema check for diagnosing the edematous form of SAM [31, 37–40]. However, the diagnostic accuracy of current MUAC cut-offs among children diagnosed as severely wasted by the criterion WHZ has been a source of concerns and controversies given the considerable variation in the overlapping between the two criteria [21, 22, 39–47]. The introduction of this approach was backed by evidence indicating that children with MUAC are a better predictor of near-term death than WHZ [48–53]. However, some experts are against the complete shift to "no Weight-for-Height" case-detection strategies, arguing that these strategies lead to many severely wasted children at high risk of near-term death remaining undiagnosed and untreated [20, 21, 42, 43]. The persistence of this controversy calls for more research on this issue.

Wasting is often classified as severe or moderate [13]. We focus only on severe wasting in this study. The WHO recommended cut-offs of WHZ<-3 Z-score or MUAC <115 mm [13] for diagnosing severe wasting, but as mentioned above most affected countries are scaling-up the MUAC (and edema)-only approach for screening and identification of SAM children and admission into treatment program [31, 37, 54]. Timor-Leste also uses this approach including in non-emergency settings (UNICEF, unpublished evaluation report). As shown in several countries, a substantial proportion of children with WHZ<-3 Z-score could be remaining undiagnosed when the MUAC (and edema)-only approach is being used. Indeed, it is likely that there is little overlap between the MUAC and WHZ criteria in Timor Leste as in many countries of the region [21, 24, 55]. As indicated above, the, two options for addressing this issue of low identification of children with WHZ<-3 Z-score in MUAC (and edema)-only programming contexts being evaluated globally include expanding MUAC cut-offs and combination of MUAC and WAZ indicators [37, 56–59]. No such evaluation has been done in Timor-Leste despite the almost countrywide implementation of the MUAC (and edema)-only approach.

This study explored the feasibility and appropriateness of expanding MUAC cut-offs and combining MUAC and WAZ indicators in the Timor Leste context. The aim was to identify the optimal MUAC cut-offs for use as the sole anthropometric criterion for maximizing the identification of children with WHZ<-3 in Timor Leste and explore the diagnostic

performances of a set of case definitions that combine MUAC and WAZ nutrition indicators. We include children under 6 months in our analyses, to add to the evidence base of identification criterion using anthropometric cut-offs for this age group. We also explore this diagnostic performance among stunted and non-stunted children, as it has been shown that up to 8% of wasted children are also stunted, and concurrence of wasting and stunting is associated with a significant increase in the risk of near-term death [7, 60].

## Materials and methods

### Study design and source of data

This was a secondary data analysis based on the 2020 Timor-Leste Food and Nutrition Survey (TLFNS), a nationally representative cross-sectional survey which was conducted between 4 June and 18 September 2020 using 2-stage cluster sampling, in which the sample was stratified by municipality [61]. The sample size, which of the survey was calculated using the SMART methodology [62]. The total target sample was 12,896 households and 9,048 children; however 12,881 households and 11,246 children were surveyed [61]. No sample size recalculation was done for our analyses; we used relevant information from all children surveyed in the original study. A copy already prepared for the analyses related to the original survey objectives containing all needed data was shared with us in an SPSS.sav file by UNICEF Timor-Leste.

### Study variables

The key variables of the data received used in our analyses included sex, age, municipality of residence and anthropometric parameters. Details on data collection procedures of these variables have been reported previously [61]. In summary, Weight, length/height, MUAC, and bilateral pitting edema were measured according to WHO-recommended methods and standards [61, 63]. Information on age, sex and the municipality of residence was obtained through the interview of the household key informant [61].

### Definitions

The WHO 2006 standards were used to calculate z-scores for anthropometric indices using the zscore06 macro for STATA [64]. Recumbent length/standing height was factored into calculating the z-score, per the macro procedure and formula. Any children with extreme values for z-scores according to commonly applied WHO flag recommendations were excluded from the analysis (Fig 1). Predefined MUAC categories were considered for this study. They were defined using different MUAC cut-offs, namely MUAC <110 mm, MUAC <115 mm, MUAC < 120 mm, MUAC <125 mm, MUAC < 130 mm, and MUAC <135 mm.

Based on the literature on MUAC accuracy for diagnosing severe wasting, we defined five age groups, namely 0 to 59 months old, 0 to 5 months old, 6 to 59 months old, 6 to 23 months old, and 24 to 59 months old [22–24, 47].

WHZ WHO classification was used to define severe wasting (WHZ<-3), which was the condition (disease) for which the diagnostic accuracy of MUAC and the different case definitions (diagnostic criteria/tests) were determined. The case definitions evaluated combined a MUAC threshold and severe underweight as defined by the standard WAZ threshold of <-3 Z-score (Table 1).

### Statistical methods

Data analysis was performed using the statistical software Stata version 16.0 (StataCorp 2019, Stata Statistical Software: Release 16. College Station, TX: StataCorp LLC). We used the Stata

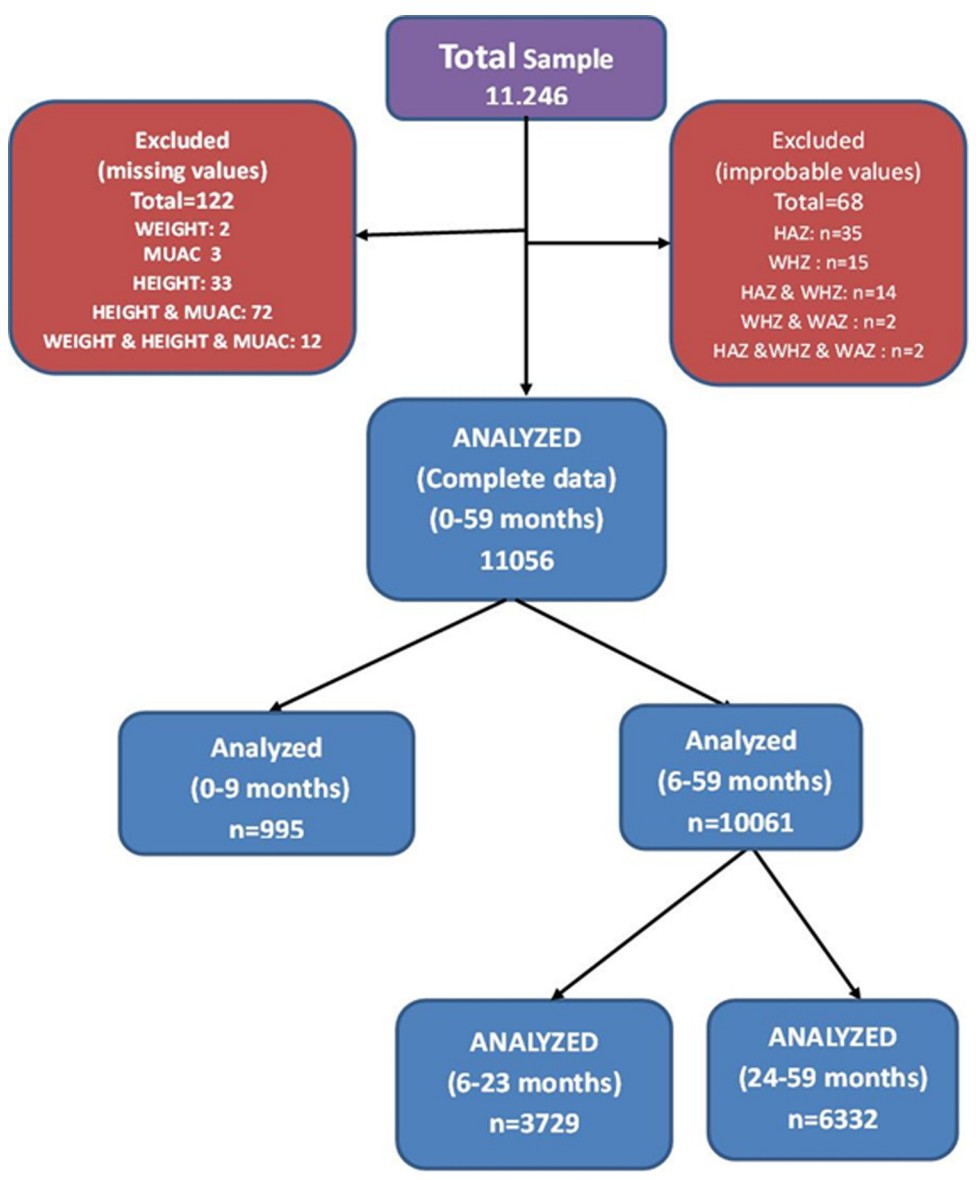

**Fig 1. Analysis inclusion.**

**Table 1. Evaluated case definitions for the identification of severe wasting.**

| Denomination | Definition of positive test |
| --- | --- |
| Case definition 1 | MUAC[1]<115 mm or WAZ[2]<-3 Z-score |
| Case definition 2 | MUAC<125 mm or WAZ<-3 Z-score |
| Case definition 3 | MUAC<130 mm or WAZ<-3 Z-score |
| Case definition 4 | MUAC<135 mm or WAZ<-3 Z-score |
| Case definition 5 | MUAC <110 mm or WAZ <-2 Z-score |

[1]MUAC = Mid-Upper Arm Circumference

[2]WAZ = Weight-for-Age Z-score

commands diagt and cutpt to determine the optimal cut-offs of MUAC, and their related accuracy characteristics, and the accuracy characteristics of the four case definitions described above. The accuracy characteristics are described in terms of sensitivity, specificity, Youden index, Area Under the receiver-operating characteristic curve (AUC), positive and negative predictive values, and positive and negative likelihood ratios. These parameters were calculated with and without disaggregating the sample per sex or age group. All these parameters are presented with their 95 percent confidence interval (95%CI). As recommended, the Youden index was used to determine the optimal MUAC cut-off for diagnosing severe wasting, with the optimal cut-off being the MUAC value having the highest Youden index value [65, 66]. The Youden index and the AUC were also used to rate the overall diagnostic performance of the optimal cut-offs and the evaluated case definitions using the cut-offs proposed in the literature [66, 67]. Based on the Youden index, the overall diagnostic accuracy is deemed insufficient, acceptable, good, very good, and excellent if its value is <55, 55–64, 65–74, 75–84, and ≥85, respectively. Based on AUC, the overall diagnostic accuracy is deemed bad, sufficient, good, very good, and excellent if its value is <0.6, 0.6-<0.7, 0.7-<0.8, 0.8-<0.9, and 0.9–1.0, respectively.

### Ethical considerations

No specific ethical approval was sought for this study as our analyses were conducted on data already collected for the assessment of prevalence of malnutrition and determination of the country's food security situation. However, we obtained an authorization to conduct the secondary analyses of the survey data from the Ministry of Health. The original survey received formal Ethical approval from the Ethics committee of the National Institute of Health of the Ministry of Health (reference #: 369/MS-INS/DE/III/2020). The dataset used for the analyses had all data anonymized. Informed written consent to participate to the original survey was obtained for all children prior to collecting the information used in our analyses.

## Results

### Description of the study population

The analyzed dataset included a total of 11,246 children aged from 0 to 59 months. After removing values flagged as improbable and observations with missing information on WHZ, WAZ, height-for-age Z-score (HAZ), or MUAC, 11,056 children (98.3% of the total) contributed to the sample used for further analyses (Fig 1). Children below 6 months of age represented 9.8% of the analyzed sample. Because 12 of the 13 municipalities surveyed were predominantly rural, three-quarters (75.6%) of the children of the sample were from rural households. Overall, boys represented 52.1% of the final sample. The predominance of boys was encountered in the 6 to 59 months age group (52.5%) but not in the below 6 months age group (48.0%). The distribution of the sex and the age groups varied from one municipality to the other, although for the sex, the boy's predominance was observed in nearly all the municipalities, and the percentage of 6–59 months old children was around 90% in all the municipalities (S1 Fig)

### Overlap of severe underweight, severe wasting, and stunting by age group

Fig 2 below depicts the overlapping of severe wasting (WHZ<-3), low MUAC (MUAC<115 mm), and severe underweight (WAZ<-3). It shows that WHZ<-3, MUAC<115 mm, and WAZ<-3 identify different children (i.e., poorly overlap) in all the age groups considered. Only a few children classified as severely wasted by WHZ<-3 also had a MUAC<115 mm,

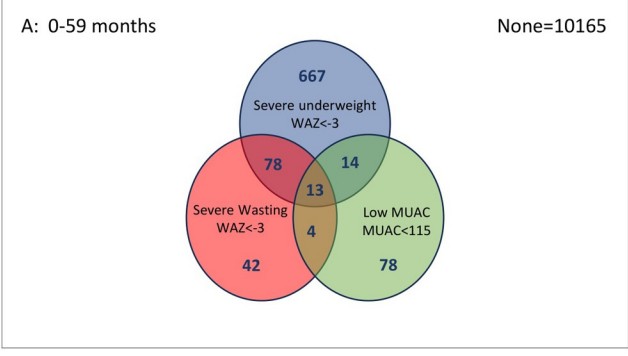
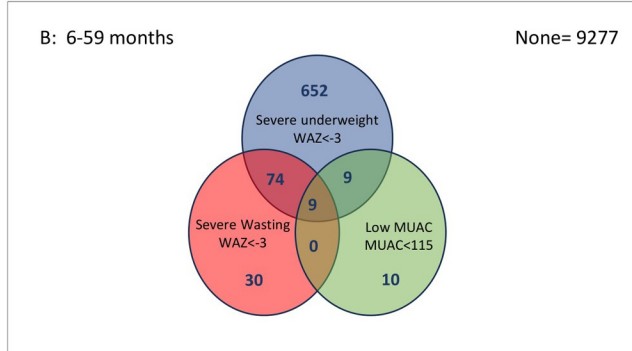
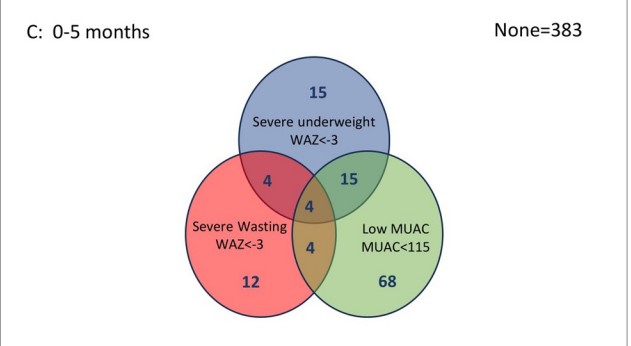
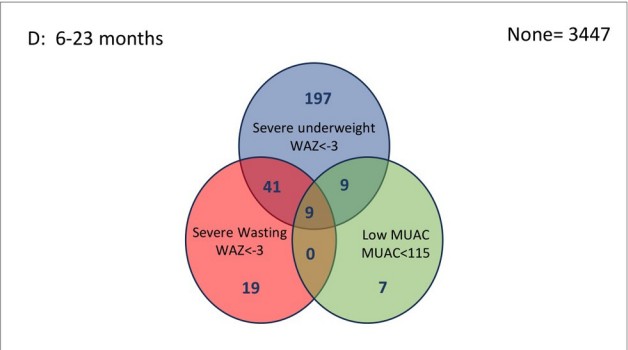

**Fig 2. Overlap between severe wasting, severe underweight, and MUAC <115 mm in different age groups.**

while over two-thirds of them had a WAZ<-3 except in the 0 to 5 months age group. In all age groups, most children with WAZ<-3 did not have either WHZ<-3 or MUAC<115 mm (Fig 2). Sixty of the 0–5 months infants had a MUAC<110 mm, of whom 11.7% (7/60) also had a WHZ<-3 (representing 29.2% of those of the age group with WHZ<-3) and 13.3% (8/60) also had a WAZ<-3 (representing 28.5% of all those of the age group with a WAZ<-3). Most of these infants (83.3%) had WHZ≥-3 and WAZ≥-3.

In terms of the overlap between stunting status and the other three indicators, overall, 45.2% (62/137) of severely wasted children also had a HAZ<-2, 94.9% _ (772/ of the severely underweight also had a HAZ<-2 and 38.5% (42/109) of those with a MUAC<115 mm also had a HAZ<-2. There was almost no overlap between severe wasting and stunting among the 0–5 month-old infants (Fig 3). The overlap between severe wasting and stunting was proportionally nearly the same (55.1% and 52.3%) for the 6–23 month and 24 to 59 month age groups, respectively (Fig 3). All the cases having both a MUAC<115 mm and a HAZ<-2 were aged less than 24 months while in all age groups, the concurrence of WAZ<-3 and HAZ<-2 was observed in over half of the children (Fig 3).

## Prevalence of different deficits

Table 2 describes the nutritional status of the study population. Regardless of the nutritional indicator used, an important percentage of children were undernourished. Stunting and underweight were more common than wasting in all considered age groups, and concurrence of wasting and stunting was more prevalent in children beyond 5 months of age.

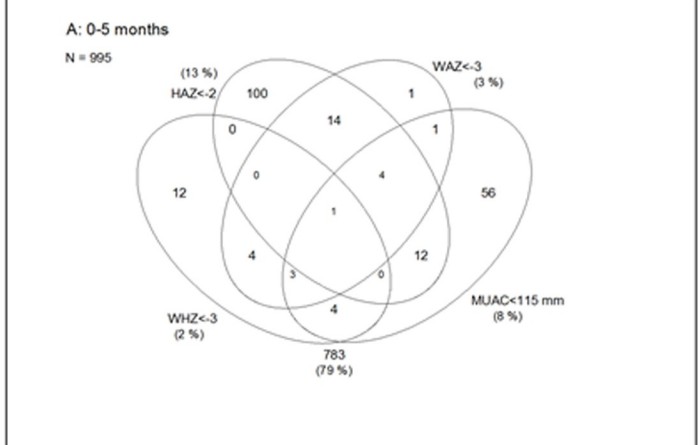
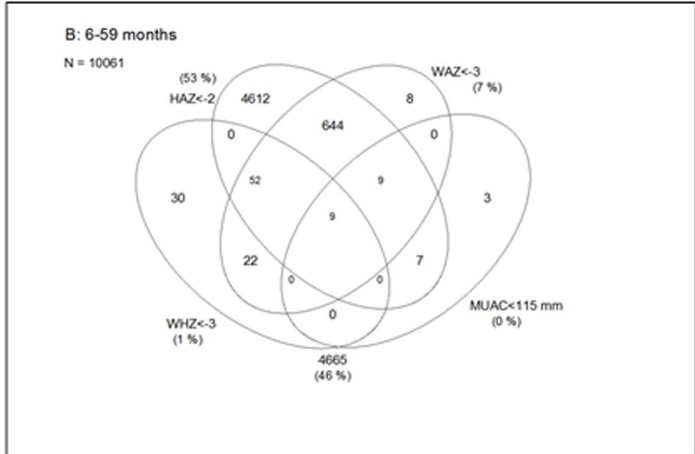
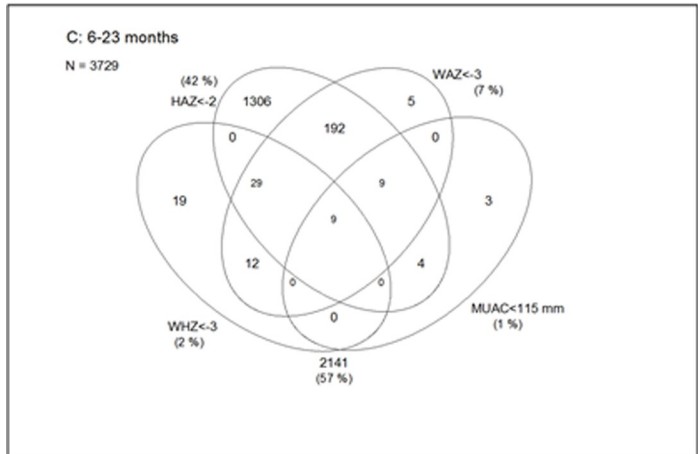
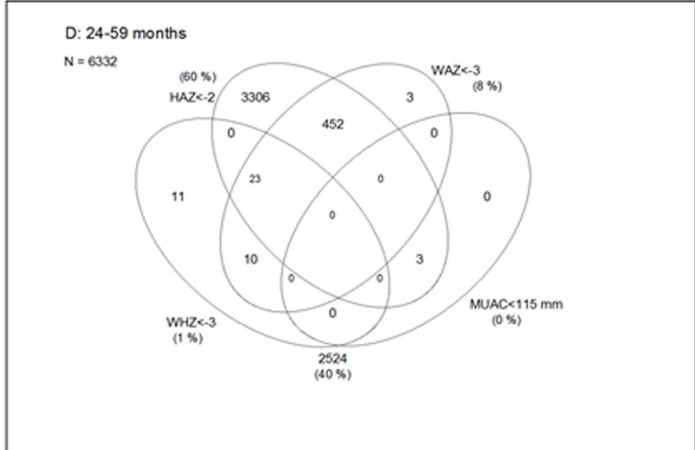

**Fig 3. Overlap between stunting, severe wasting, severe underweight, and MUAC <11.5 cm in different age groups.**

## Using Mid-Upper Arm Circumference cut-off to identify severe wasting diagnosed by low weight-for-height

The results of the analysis that evaluated the diagnostic accuracy for diagnosing severe wasting of all the MUAC values recorded showed that the optimal MUAC cut-off increased with age from 129 mm in the age group 0 to 5 months old to 142 mm in the age group 24–59 months old (Table 3). For the five age groups considered, the optimal cut-off was > 125 mm, including in infants of 0 to 5 months age group (Table 3). The AUC values were between 0.70 and 0.80 while the Youden index values were below 55% (Table 3). The gender-specific optimal MUAC cut-offs for diagnosing severe wasting also increased with age and were higher in boys than girls (Table 3). Generally, the gender-specific optimal cut-off had better diagnostic characteristics than those calculated for both sex (Table 3).

## Diagnostic accuracy of selected Mid-Upper Arm Circumference cut-offs

Table 4 summarizes diagnostic accuracies of selected MUAC cut-offs in diagnosing severe wasting (WHZ<-3). Based on the Youden Index and the AUC, 135 mm cut-off was the most accurate for the age groups 6 to 59 months and 0 to 59 months, but 130 mm cut-off was the most accurate

**Table 2. Nutritional status of the study population by different anthropometric indicators.**

| Nutritional Indicator | Age groups | | | | | | | |
|---|---|---|---|---|---|---|---|---|
| | 0–59 months | | 0–5 months | | 6–23 months | | 6–59 months | |
| | n | % | N | % | N | % | N | % |
| MUAC[1] (mm) | | | | | | | | |
| ≥135 | 8676 | 78.47 | 435 | 43.72 | 2504 | 67.15 | 8241 | 81.91 |
| 130–134 | 1271 | 11.50 | 171 | 17.19 | 695 | 18.64 | 1100 | 10.93 |
| 125–129 | 644 | 5.42 | 162 | 16.28 | 340 | 9.12 | 482 | 4.79 |
| 120–124 | 251 | 2.27 | 86 | 8.64 | 129 | 3.46 | 165 | 1.64 |
| 115–119 | 105 | 0.95 | 60 | 6.03 | 36 | 0.97 | 45 | 0.45 |
| 110–114 | 38 | 0.34 | 21 | 2.11 | 14 | 0.38 | 17 | 0.17 |
| <110 | 71 | 0.64 | 60 | 0.63 | 11 | 0.29 | 11 | 0.11 |
| Total | 11056 | 100.0 | 995 | 100.0 | 3729 | 100.0 | 10061 | 100.0 |
| WHZ[2] (Z-score) | | | | | | | | |
| ≥-2.0 | 10200 | 92.33 | 919 | 92.36 | 3372 | 90.43 | 9289 | 92.33 |
| ≥-3.0 -<-2.0 | 711 | 6.43 | 52 | 5.23 | 288 | 7.72 | 659 | 6.55 |
| <-3.0 | 138 | 1.24 | 24 | 2.41 | 69 | 1.85 | 113 | 1.12 |
| Total | 11056 | 100.0 | 995 | 100.0 | 3729 | 100.0 | 10061 | 100.0 |
| WAZ[3] (Z-score) | | | | | | | | |
| ≥-2.0 | 7470 | 67.57 | 890 | 89.45 | 2703 | 72.49 | 6580 | 65.40 |
| ≥-3.0 -<-2.0 | 2814 | 25.45 | 77 | 7.74 | 770 | 20.65 | 2727 | 27.20 |
| <-3.0 | 772 | 6.98 | 28 | 2.81 | 256 | 6.86 | 744 | 7.39 |
| Total | 11056 | 100.0 | 995 | 100.0 | 3729 | 100.0 | 10061 | 100.0 |
| Acute malnutrition[4] | | | | | | | | |
| None | 9908 | 89.62 | 728 | 73.17 | 3282 | 88.01 | 9180 | 91.24 |
| Moderate | 919 | 8.31 | 170 | 17.09 | 362 | 9.71 | 749 | 7.44 |
| Severe | 229 | 2.07 | 97 | 9.74 | 85 | 2.28 | 132 | 1.31 |
| Total | 11056 | 100.0 | 995 | 100.0 | 3729 | 100.0 | 10061 | 100.0 |
| Wasting & stunting group[5] | | | | | | | | |
| Not wasted & no stunted | 5202 | 47.05 | 795 | 79.90 | 2022 | 54.22 | 4407 | 43.80 |
| Wasted only | 390 | 3.53 | 69 | 6.93 | 158 | 4.24 | 321 | 3.19 |
| Stunted only | 5006 | 45.28 | 124 | 12.46 | 1350 | 36.20 | 4882 | 48.52 |
| Wasted & stunted | 458 | 4.14 | 7 | 0.70 | 199 | 5.34 | 451 | 4.48 |
| Total | 11056 | 100.0 | 995 | 100.0 | 3729 | 100.0 | 10061 | 100.0 |

[1]MUAC = mid-upper arm circumference; [2]WHZ = weight-for-height Z-score; [3]WAZ = weight-for-age Z-score; [4]Acute malnutrition = categories based on WHO case definition using weight-for-height Z-score, mid-upper arm circumference and presence of bilateral pitting edema as independent indicators; [5]The definition and categorization are based on weight-for-height Z-score and height-for-age Z-score indices only and do not consider the value of mid-upper arm circumference.

in the age groups 0 to 5 months and 6 to 23 months (Table 4). Overall, both the 135 mm and the 130 mm cut-offs had insufficient diagnostic accuracy in all the age groups, with sensitivity in children 24–59 months being particularly low (Table 4). The 135 mm was the closest to the optimal cut-offs of the different age groups (Table 4). The cut-off 110 mm was evaluated only in the age group 0 to 5 months, and it had insufficient accuracy in this age group (Table 4).

## Case definitions combining Mid-Upper Arm Circumference and Weight-for-Age Z-score

Table 5 presents the diagnostic accuracy parameters of several case definitions for identifying severe wasting in under-five children, combining MUAC cut-offs and WAZ cut-offs. Among

**Table 3. Optimal Mid-Upper Arm Circumference cut-off for the diagnosis of severe wasting in children under five years and their corresponding accuracy characteristics by age groups and gender.**

| Age group | Sample | n positive cases | Prevalence | Optimal MUAC Cutoff (mm) | Sensitivity | Specificity | Youden Index (%) | Correctly Classified | LR(+)[1] | LR(-)[2] | AUC[3] |
|---|---|---|---|---|---|---|---|---|---|---|---|
| **All children** | | | | | | | | | | | |
| 0–5 months | 995 | 24 | 2.41 | < 129 | 75.00% | 65.19% | 40.19 | 65.43% | 2.6076 | 0.4641 | 0.701 |
| 6–23 months | 3729 | 69 | 1.85 | < 134 | 78.26% | 73.85% | 52.11 | 73.93% | 3.3972 | 0.3341 | 0.761 |
| 24–59 months | 6332 | 44 | 0.69 | < 142 | 74.75% | 72.73% | 47.48 | 74.73% | 2.7407 | 0.3472 | 0.737 |
| 6–59 months | 10061 | 113 | 1.12 | < 136 | 80.45% | 70.80% | 51.25 | 80.34% | 2.7547 | 0.2762 | 0.756 |
| 0–59 months | 11056 | 137 | 1.24 | < 134 | 68.61% | 82.59% | 51.20 | 82.42% | 2.6314 | 0.2537 | 0.756 |
| **Boys** | | | | | | | | | | | |
| 0–5 months | 489 | 11 | 2.25 | <127 | 72.73% | 79.29% | 52.02 | 79.14% | 2.9073 | 0.2848 | 0.760 |
| 6–23 months | 1948 | 44 | 2.26 | <134 | 77.27% | 78.31% | 55.58 | 78.29% | 3.4456 | 0.2807 | 0.778 |
| 24–59 months | 3335 | 26 | 2.26 | <142 | 69.23% | 76.85% | 46.08 | 76.79% | 2.4977 | 0.3344 | 0.730 |
| 6–59 months | 5283 | 70 | 1.33 | <139 | 77.14% | 74.35% | 51.49 | 74.39% | 3.2529 | 0.3325 | 0.757 |
| 0–59 months | 5772 | 81 | 1.40 | <140 | 82.72% | 68.34% | 51.06 | 68.54% | 3.9537 | 0.3828 | 0.755 |
| **Girls** | | | | | | | | | | | |
| 0–5 months | 506 | 13 | 2.57 | <117 | 61.54% | 87.83% | 49.37 | 87.15% | 2.2836 | 0.1978 | 0.747 |
| 6–23 months | 1781 | 25 | 1.40 | <130 | 72.00% | 82.40% | 54.40 | 82.26% | 2.943 | 0.2444 | 0.772 |
| 24–59 months | 2997 | 18 | 0.60 | <135 | 66.67% | 90.37% | 57.04 | 90.22% | 2.711 | 0.1445 | 0.785 |
| 6–59 months | 4778 | 43 | 0.90 | <136 | 76.74% | 77.97% | 54.71 | 77.96% | 3.3528 | 0.287 | 0.774 |
| 0–59 months | 5284 | 56 | 1.06 | <134 | 75.00% | 79.65% | 54.65 | 79.60% | 3.1859 | 0.2714 | 0.773 |

[1] LR (+) = Positive Likelihood Ratio; [2] LR (-) = Negative Likelihood Ratio; [3] AUC = Area Under the Curve (ROC area)

infants aged 0 to 5 months, the MUAC<110 mm or WAZ<-2 case definition had the most acceptable diagnostic accuracy values, with the Youden index being above 55%. This case definition classified 15.5 (95% CI: 13.3–17.9) % of the infants of this age group as eligible for intervention. In all other age groups considered, the case definition combining the MUAC cut-off 130 mm and WAZ<-3 was slightly more accurate than the other case definitions evaluated. The AUC ranged from 0.834 to 0.847, while the Youden index ranged from 64.6% to 69.4%. This case definition classified as eligible for intervention (SAM treatment), 14.4 (95%CI: 13.7–15.1) % and 11.9 (95%CI: 11.3–12.5) % of 0 to 59 months old and 6 to 59 months old children, respectively. In children 6 to 5 months who were eligible for intervention (SAM treatment) based on the case definition MUAC<130 mm or WAZ<-3, it was found that 29.7 (95%CI: 27.1–32.3) % were not either wasted or severely underweight. In children 0 to 5 months of age who were eligible for nutrition-specific intervention based on the case definition MUAC< 110 mm or WAZ<-2, it was found that 72.7 (95%CI: 65.0–79.6) % were not wasted by the criterion WHZ<-2.

The case definition 'MUAC<130 mm or WAZ<-3 performed better in boys than in girls and in stunted children than non-stunted children (Table 6). The level of accuracy was good to very good in both sexes and for all the age groups considered, with the AUC varying between 0.805 and 0.865 and the Youden index varying between 61.0% and 73.1% (Table 6). The accuracy of this case definition was the highest among children who were stunted, among whom it was found to be very good or excellent according to the metric used (AUC or Youden index) and the age group considered (Table 6). Interestingly, with this case definition, 80.3 (95% CI:76.3–83.8) % of children 6 to 59 months old concurrently wasted and stunted were considered eligible for SAM treatment while this percentage was 13.5 (95%CI:10.5–17.3) % and 3.8 (95%CI:2.2–6.0) % when this eligibility was defined based on current WHZ and MUAC-based case definitions of SAM, respectively.

**Table 4. Diagnostic accuracy of selected practical cut-offs of Mid-Upper Arm Circumference for the diagnosis of severe wasting.**

| Age group | MUAC Cut-off (mm) | Sensitivity | Specificity | LR(+)[1] | LR(-)[2] | Youden Index (%) | AUC[3] | Correctly Classified |
|---|---|---|---|---|---|---|---|---|
| 0–5 months (n = 995) | <235 | 100.00% | 0.00% | 1 | - | 0 | 0.500 | 2.41% |
| | <135 | 87.50% | 44.49% | 1.5763 | 0.281 | 31.99 | 0.660 | 45.53% |
| | <130 | 75.00% | 61.79% | 1.9629 | 0.4046 | 36.79 | 0.684 | 62.11% |
| | <125 | 54.17% | 77.96% | 2.4577 | 0.5879 | 32.13 | 0.661 | 77.39% |
| | <120 | 41.67% | 86.51% | 3.0884 | 0.6743 | 28.18 | 0.641 | 85.43% |
| | <115 | 33.33% | 92.48% | 4.4338 | 0.7209 | 25.81 | 0.629 | 91.06% |
| | <110 | 29.17% | 94.54% | 5.3436 | 0.7492 | 23.71 | 0.619 | 92.96% |
| 6–23 months (n = 3729) | <235 | 100.00% | 0.00% | 1 | - | 0 | 0.500 | 1.85% |
| | <135 | 79.71% | 68.03% | 2.4935 | 0.2982 | 47.74 | 0.739 | 68.25% |
| | <130 | 63.77% | 86.72% | 4.8023 | 0.4178 | 50.49 | 0.752 | 86.30% |
| | <125 | 44.93% | 95.66% | 10.3418 | 0.5757 | 40.59 | 0.703 | 94.72% |
| | <120 | 26.09% | 98.83% | 22.2043 | 0.7479 | 24.92 | 0.625 | 97.48% |
| | <115 | 13.04% | 99.56% | 29.8371 | 0.8734 | 12.6 | 0.563 | 97.96% |
| | <110 | 5.80% | 99.81% | 30.3099 | 0.9438 | 5.61 | 0.528 | 98.07% |
| 24–59 months (6332) | <235 | 100.00% | 0.00% | 1 | - | 0 | 0.500 | 0.69% |
| | <135 | 50.00% | 90.89% | 5.4869 | 0.5501 | 40.89 | 0.704 | 90.60% |
| | <130 | 22.73% | 97.14% | 7.9394 | 0.7955 | 19.87 | 0.599 | 96.62% |
| | <125 | 6.82% | 99.28% | 9.5272 | 0.9385 | 6.1 | 0.531 | 98.64% |
| | <120 | 2.27% | 99.83% | 12.9919 | 0.979 | 2.1 | 0.511 | 99.15% |
| | <115 | 0.00% | 99.95% | 0 | 1.0005 | -0.05 | 0.500 | 99.26% |
| | <110 | 0.00% | 100.00% | | 1 | 0 | 0.500 | 99.31% |
| 6–59 months (n = 10061) | <235 | 100.00% | 0.00% | 1 | - | 0 | 0.500 | 1.12% |
| | <135 | 68.14% | 82.48% | 3.8891 | 0.3863 | 50.62 | 0.753 | 82.32% |
| | <130 | 47.79% | 93.31% | 7.138 | 0.5596 | 41.1 | 0.706 | 92.79% |
| | <125 | 30.09% | 97.95% | 14.6726 | 0.7138 | 28.04 | 0.640 | 97.19% |
| | <120 | 16.81% | 99.46% | 30.9754 | 0.8364 | 16.27 | 0.581 | 98.53% |
| | <115 | 7.96% | 99.81% | 41.7007 | 0.9221 | 7.77 | 0.539 | 98.78% |
| | <110 | 3.54% | 99.93% | 50.3059 | 0.9653 | 3.47 | 0.517 | 98.85% |
| 0–59 months (n = 11056) | <235 | 100.00% | 0.00% | 1 | - | 0 | 0.500 | 1.24% |
| | <135 | 71.53% | 79.10% | 3.4227 | 0.3599 | 50.63 | 0.753 | 79.01% |
| | <130 | 52.55% | 90.50% | 5.5337 | 0.5242 | 43.05 | 0.715 | 90.03% |
| | <125 | 34.31% | 96.17% | 8.9616 | 0.6831 | 30.48 | 0.652 | 95.41% |
| | <120 | 21.17% | 98.31% | 12.4937 | 0.8019 | 19.48 | 0.597 | 97.35% |
| | <115 | 12.41% | 99.16% | 14.7273 | 0.8834 | 11.57 | 0.558 | 98.08% |
| | <110 | 8.03% | 99.45% | 14.6118 | 0.9248 | 7.48 | 0.537 | 98.32% |

[1] LR (+) = Positive Likelihood Ratio; [2] LR (-) = Negative Likelihood Ratio; [3] AUC = Area Under the Curve

## Discussion

In this study, we aimed to identify the optimal MUAC cut-off for Timor-Leste in under-five children and evaluate the diagnostic accuracy of various MUAC cut-offs when used as the sole diagnostic criterion and when combined with WAZ, the indicator for underweight. We looked at the overall diagnostic accuracy and by child age groups, gender and whether or not they were stunted. We used a nationally representative sample of under-five children collected during the 2020 national food and nutrition survey [61]. The results of our analyses show that the optimal MUAC cut-off for identifying children with WHZ<-3 in all the age groups considered was much higher than the currently recommended cut-off of 115 mm for children aged 6 to 59

**Table 5. Diagnostic test accuracy parameters for various case definitions combining different mid-upper arm circumference cut-offs and the indicator of severe underweight in diagnosing severe wasting among underfive children.**

| Age group | Positive test | sensitivity (%) | Specificity (%) | AUC[1] | Youden index (%) | PPV[2] (%) | NPV[3] (%) | LR (+)[4] | LR (-)[5] | Odds ratio |
|---|---|---|---|---|---|---|---|---|---|---|
| 0–5 months (n = 995 & prevalence = 2.41%) | MUAC[5]<110 mm or WAZ[6]<-2 | 75.0 | 86.0 | 0.805 | 61.0 | 11.7 | 99.3 | 5.35 | 0.291 | 18.4 |
| | MUAC<115 mm or WAZ<-3 | 50.0 | 90.9 | 0.705 | 40.9 | 12.0 | 98.7 | 5.52 | 0.55 | 10.0 |
| | MUAC<125 mm or WAZ<-3 | 66.7 | 77.0 | 0.719 | 43.7 | 6.7 | 98.8 | 2.9 | 0.43 | 6.7 |
| | MUAC<130 mm or WAZ<-3 | 83.3 | 61.3 | 0.723 | 44.6 | 5.1 | 99.3 | 2.15 | 0.27 | 7.9 |
| | MUAC<135 mm or WAZ<-3 | 87.5 | 44.4 | 0.659 | 31.9 | 3.7 | 99.3 | 1.57 | 0.28 | 5.6 |
| 6–23 months (n = 3729 & prevalence = 1.85%) | MUAC<115 mm or WAZ<-3 | 72.5 | 94.2 | 0.833 | 66.7 | 19.0 | 99.5 | 12.5 | 0.29 | 42.6 |
| | MUAC<125 mm or WAZ<-3 | 73.9 | 91.6 | 0.827 | 65.5 | 14.2 | 99.5 | 8.78 | 0.29 | 30.8 |
| | MUAC<130 mm or WAZ<-3 | 82.6 | 84.3 | 0.834 | 66.9 | 9.0 | 99.6 | 5.25 | 0.206 | 25.4 |
| | MUAC<135 mm or WAZ<-3 | 89.9 | 67.2 | 0.735 | 57.1 | 4.9 | 99.7 | 2.74 | 0.15 | 18.1 |
| 24–59 months (n = 6332 & prevalence = 0.69%) | MUAC<115 mm or WAZ<-3 | 75.0 | 92.7 | 0.839 | 67.7 | 6.7 | 99.8 | 10.3 | 0.27 | 38.2 |
| | MUAC<125 mm or WAZ<-3 | 77.3 | 92.6 | 0.849 | 69.9 | 6.8 | 99.8 | 10.4 | 0.25 | 42.3 |
| | MUAC<130 mm or WAZ<-3 | 77.6 | 91.6 | 0.844 | 69.2 | 6.0 | 99.8 | 9.17 | 0.25 | 36.9 |
| | MUAC<135 mm or WAZ<-3 | 81.8 | 87.2 | 0.845 | 69.0 | 4.3 | 99.9 | 6.38 | 0.21 | 30.6 |
| 6–59 months (n = 10061 & prevalence = 1.12%) | MUAC<115 mm or WAZ<-3 | 73.5 | 93.3 | 0.834 | 66.8 | 11.0 | 99.7 | 10.9 | 0.28 | 38.3 |
| | MUAC<125 mm or WAZ<-3 | 75.2 | 92.2 | 0.837 | 67.4 | 9.9 | 99.7 | 9.64 | 0.27 | 35.9 |
| | MUAC<130 mm or WAZ<-3 | 80.5 | 88.9 | 0.847 | 69.4 | 7.6 | 99.8 | 7.24 | 0.21 | 34.4 |
| | MUAC<135 mm or WAZ<-3 | 86.7 | 79.8 | 0.833 | 66.5 | 4.6 | 99.8 | 4.3 | 0.17 | 25.8 |
| 0–59 months (n = 11056 & prevalence = 1.24%) | MUAC<115 mm or WAZ<-3 | 69.3 | 93 | 0.812 | 62.3 | 11.1 | 99.6 | 9.98 | 0.33 | 30.3 |
| | MUAC<125 mm or WAZ<-3 | 73.7 | 90.9 | 0.823 | 64.6 | 9.2 | 99.6 | 8.06 | 0.29 | 27.9 |
| | MUAC<130 mm or WAZ<-3 | 81.0 | 86.4 | 0.837 | 67.4 | 7.0 | 99.7 | 5.97 | 0.22 | 27.2 |
| | MUAC<135 mm or WAZ<-3 | 86.9 | 76.7 | 0.832 | 63.6 | 4.5 | 99.8 | 3.72 | 0.17 | 21.7 |

[1]AUC = Area under the curve (ROC area); [2]PPV = Positive Predictive Value; [3]NPV = Negative Predictive Value; [4]LR(+) = Positive Likelihood Ratio; [5]LR(-) = Negative Likelihood Ratio; [5] MUAC = Mid Upper Arm Circumference; [6]WAZ = weight-for-Age

months and 110 mm for infants younger than 6 months and even higher than that of 125 mm used for the identification of both moderate and severe wasting [10, 13]. The level of diagnostic accuracy of these cut-offs (117 mm to 142 mm according to gender and age) and that of all the alternative cut-offs tested remained sub-optimal as the Youden index values were always less than or around 55%, the threshold for considering a diagnostic test useful [68, 69]. A combination of MUAC and WAZ optimized the identification of children with WHZ<-3, and among the combination tested, the combination of MUAC<130 mm and WAZ<-3 Z-score had the best diagnostic accuracy in all the age groups considered for children aged between 6 and 59 months while the combination MUAC<110 mm and WAZ<-2 had the best diagnostic characteristics in infants of the 0 to 5 months age group. These are important findings that will greatly contribute to local and global evidence on how to ensure early identification of severe wasting among vulnerable children and expand treatment coverage.

It is well documented that these MUAC and WHZ currently recommended by WHO for the diagnostic of wasting poorly overlap and identify different children and that it is challenging to use the WHZ indicator in community-based nutrition interventions and in resources-constrained contexts [13, 21, 37, 45, 70–72]. Because of the difficulties related to the use of WHZ, it has been challenging to consistently use both WHZ and MUAC in the field. This situation has led, many wasting treatment programs to shift to a MUAC (and edema)-only approach, a protocol simplification that uses MUAC as the sole anthropometric indicator for detecting and admitting wasted children and edema check for the detection of those with

**Table 6. Diagnostic accuracy of the case definition combining Mid-Upper Arm Circumference below 130 millimeters and weight-for-age Z-score below minus 3 Z-score in diagnosing severe wasting among under-five children according to sex and stuntedness.**

| Category | sample | prevalence | sensitivity (%) | Specificity (%) | AUC[1] | Youden index | PPV[2] (%) | NPV[3] (%) | LR (+)[4] | LR (-)[5] | Odds ratio |
|---|---|---|---|---|---|---|---|---|---|---|---|
| **Boys** | | | | | | | | | | | |
| 0–59 months | 5772 | 1.40 | 82.7 | 88.5 | 0.856 | 71.2 | 9.3 | 99.7 | 7.21 | 0.195 | 36.9 |
| 6–59 months | 5283 | 1.33 | 82.9 | 90.2 | 0.865 | 73.1 | 10.2 | 99.7 | 8.44 | 0.19 | 44.4 |
| 6–23 months | 1948 | 2.26 | 84.1 | 87.3 | 0.857 | 71.4 | 13.3 | 99.6 | 6.62 | 0.182 | 36.3 |
| **Girls** | | | | | | | | | | | |
| 0–59 months | 5284 | 1.06 | 78.6 | 84.1 | 0.814 | 62.7 | 5.0 | 99.7 | 4.96 | 0.255 | 19.5 |
| 6–59 months | 4778 | 0.90 | 76.7 | 87.5 | 0.821 | 64.2 | 5.3 | 99.8 | 6.12 | 0.266 | 23.0 |
| 6–23 months | 1781 | 1.40 | 80.0 | 81.0 | 0.805 | 61.0 | 5.6 | 99.6 | 4.21 | 0.2 | 17.0 |
| **Non-stunted** | | | | | | | | | | | |
| 0–59 months | 5592 | 1.34 | 65.3 | 90.6 | 0.780 | 55.9 | 8.6 | 99.5 | 6.96 | 0.383 | 18.3 |
| 6–59 months | 4728 | 1.10 | 57.7 | 95.4 | 0.766 | 53.1 | 12.3 | 99.5 | 12.7 | 0.443 | 28.6 |
| 6–23 months | 2180 | 1.42 | 61.3 | 91.3 | 0.763 | 52.6 | 9.2 | 99.4 | 7.04 | 0.424 | 16.6 |
| **Stunted** | | | | | | | | | | | |
| 0–59 months | 5464 | 1.13 | 100.0 | 82.2 | 0.911 | 82.2 | 6.0 | 100.0 | 5.60 | 0.0 | - |
| 6–59 months | 5333 | 1.14 | 100.0 | 83.3 | 0.915 | 83.3 | 6.4 | 100.0 | 5.90 | 0.0 | - |
| 6–23 months | 1549 | 2.45 | 100.0 | 74.3 | 0.871 | 74.3 | 8.9 | 100.0 | 3.88 | 0.0 | - |

[1]AUC = Area under the curve (ROC area); [2]PPV = Positive Predictive Value; [3]NPV = Negative Predictive Value; [4]LR (+) = Positive Likelihood Ratio; [5]LR (-) = Negative Likelihood Ratio

edematous malnutrition [37, 70, 71]. Although there is a consensus that MUAC is a good predictor of mortality, some experts are concerned about the severely wasted children that are undetected when the MUAC (and edema)-only approach is used [20, 50]. The proportion of undetected wasted children varies from -country to country but can be very large in some high-burden countries; hence, the need to identify approaches that can allow to increase the number of children severely wasted by the WHZ criterion [20]. A mitigation strategy already being used is to expand the MUAC cut-offs using 125 mm instead of 115 mm, but there is not yet a consensus on the benefit of such strategy [20, 21, 24, 37, 73, 74]. Indeed, the diagnostic accuracy of such shift remains insufficient.

It has been shown that in many South and Southeast Asian countries, the overlap between MUAC and WHZ, as per the current WHO criteria for severe wasting, can be very poor [21, 27, 75–77]. There have been attempts to address this poor concordance by determining and proposing country-specific, or even local, optimal cut-offs. Indeed, several research teams of the region embarked on studies to determine the local or country specific optimal MUAC cut-off that maximizes the accuracy in detecting children with WHZ<-3 of MUAC. The optimal MUAC cut-off identified based on community-based cross-sectional surveys for the 6–59 month age group reported in the reviewed literature, varied across studies from 125 mm to 135 mm, with variation observed even in the same country [22, 23, 75, 78, 79] For instance, the three studies conducted in India that we reviewed yielded different optimal cut-offs of 128 mm for children recruited in the villages of Melghat block of Amravati district of Maharashtra state [75], of 130 mm for Indian children from Wardha district of Maharashtrain state [78], and of 135 mm among Indian children from Meerut District of Pradesh state [23]. The optimal MUAC cut-off from studies in other Asian countries was 135 mm in Vietnamese children from Northern Midlands and mountainous areas [22], and 125 mm for Nepali children recruited in five surveys conducted in different provinces between 2011 and 2013 [79]. Facility-based cross-sectional studies have also been conducted in some Asian countries for the

same purpose and the calculated optimal cut-offs were 140 mm in Indonesian children from Central Java recruited at different rural primary health facilities [80], and 127 mm for Karachi children (Pakistan) assessed at the largest tertiary hospital of the town [81]. In comparison to the evidence from the reviewed studies cited above, the optimal cut-off of 136 mm we observed for children 6–59 months is within the range of previously reported cut-offs and close to the frequently reported cut-off for the 6–59 months old children of 135 mm [24, 82]. Nonetheless, our findings and that from the other cited studies back the need for defining context specific cut-offs for use in areas applying the MUAC-only programming approach for recruiting and monitoring children with severe wasting. They are in favor of contextualizing the MUAC case definition for SAM for Timor-Leste.

Evidence from the only two published studies from the African continent confirmed the challenge of defining a global or even a continental unique MUAC cut-off for detecting children with WHZ<-3 [83, 84]. Indeed, a study conducted in Ethiopia that used pooled data from nutrition surveys carried out in 49 different districts between 2016 and 2019, found that 133 mm was the most accurate MUAC cut-off for identifying children with WHZ<-3 in the 6–59 months age group [83] while the secondary data analysis of data from the 2015 Mauritania national nutrition survey found that cut-off to be 138 mm for Mauritanian children of the same age group [84]. The observed variation is within the range observed in community-based Asian studies listed above.

The current practice is to use a single MUAC cut-off for children aged 6–59 months [13, 15, 44, 45]. The magnitude of the difference in the calculated optimal MUAC cut-off between the 0–5 month, 6–23 month and 24–59 month age groups of 5 mm to 13 mm echoes what has been observed in other studies of up to 15 mm difference between the optimal MUAC cut-off of the 6–23 months and the 24–59 month age groups, and supports the suggestion from some nutritionists of dividing the 0–59 month age group into smaller age groups, and define a specific cut-offs for each of them [22, 44, 45, 81–86]. Suggestions to define sex-specific cut-offs have also been made. Our data did not support such a recommendation for the Timor-Leste context, given the small magnitude of the effect of gender on the value of optimal MUAC cut-off [44, 83]. The absence of gender effect was also reported in a study conducted in Southern region of Ethiopia [87].

There has been an attempt to determine a global MUAC optimal cut-off for under-five children through meta-analysis statistical methods. Indeed, a recently published meta-analysis summarized the findings of some studies (n = 11) conducted in Asia or Africa [24]. Unfortunately, this meta-analysis omitted some of relevant studies already available in the literature including many of those we have discussed above, hence the calculated pooled estimates need to be interpreted with caution [82, 83, 86]. Also, the heterogeneity of the included studies in terms of age groups, and study design (community-based or facility-based), need to be considered when interpreting the obtained pooled cut-off [24]. Nevertheless, the meta-analysis gave some indications on the magnitude of the difference between the current recommended MUAC cut-off for the diagnosis of severe wasting. It found a pooled optimal cut-off to identify children with WHZ<-3 of 132 mm [24]. Interestingly, this value slightly falls below the 136 mm and 134 mm obtained in our study for the 6 to 59 month (age group for 7 of 11 included studies) and 0 to 59 month (age group for 3 of 11 included studies) age groups, respectively.

The optimal cut-offs reported in papers reviewed above, including the meta-analysis, and in our study are much higher to the currently recommended cut-off of 115 mm [10, 13]. Also, while these higher cut-offs increase diagnostic sensitivities, their effect on the Youden Index, which reflects overall diagnostic accuracy, varies. Some of these optimal cut-offs only achieved insufficient or marginally acceptable accuracy ratings. Consistent with what was reported in many of the reviewed studies, in our study population too, the obtained optimal cut-offs still

had insufficient accuracy, as reflected by the Youden index remaining below 55%, although there was an increase for 6–59 months from 7.8% to 50.6% [22, 79, 83, 85, 86]. Nonetheless, other studies have shown that expanding the MUAC cut-off substantially improves MUAC diagnostic accuracy. This improvement was demonstrated in two of the three Indian studies reviewed including in Meerut District (22.9% to 73.0%) and in the Indian district of Wardha (23.2% to 67.3%) [23, 78]. The Cambodian study described earlier in this paper also reported a substantial increase in the Youden Index (5.8% to 64.2%) [45]. It is worth mentioning that all these studies showed that the gain in sensitivity resulting from the expansion of the cut-offs (≥75% achieved in our study) occurs at the expense of the specificity which may result in a substantial increase in caseload in high burden countries as demonstrated in one of the Indian studies reviewed [23].

It has been shown that the level of overlapping of current MUAC and WHZ criteria for the diagnosis of severe wasting varies greatly across settings, yet comprehensive explanation of this variability remains elusive [19]. While the body shape, indirectly measured with the sitting and standing height ratio, explains some of the variations, it is insufficient to adequately explain all the differences and similarities across continents [19, 88, 89]. As previously discussed above, our findings and previously published evidence on optimal MUAC cut-off for identifying children with WHZ<-3 support the need for contextualizing these cut-offs. However, this contextualization is likely to complicate rather than simplify the development of policies and protocols for the management of SAM [22–24, 75, 83, 90].

Notably, the optimal cut-offs observed in many of the studies, regardless of the continent, are closely aligned, particularly those for the single 6 to 59 month age group [24, 82]. 136 mm. Indeed, the ranges reported for the Asian and African studies largely overlap suggesting that it is possible to identify a common cut-off for these two continents, which host the great majority of children with severe wasting [4]. Based on information from the reviewed studies, the common cut-off should be between 128 mm and 136 mm, given that most of the optimal cut-offs fall within this interval [24, 45, 75, 82, 83, 91]. Such a cut-off would be higher than the 120 mm and 125 mm that has been previously proposed and is likely to detect all children at high risk of near-term death in all contexts, as shown by one pooled analysis of four datasets of community-based prospective cohorts that showed that increasing MUAC cut-offs substantially improves the capability of detecting these deaths [92]. Also, the cut-offs for the increase in the risk of near-term death in the community reported for Bangladesh, Malawi, Senegal and Uganda confirm that the shift to such a cut-off would likely contribute to the global efforts to reducing under-five mortality [41, 57, 93]. The cut-off should be chosen taking into account the epidemiology of wasting in terms of the most affected age group and sex predominance. Our study and some of previously published revealed that the optimal cut-offs may differ according to age and sex [44, 72, 83, 94].

As previously discussed, the overall diagnostic performance of the national and local optimal MUAC cut-off for identifying children with WHZ<-3 when used as sole anthropometric indicator will remain sub-optimal in many settings. There have been suggestions to address this issue by combining MUAC and WAZ, an anthropometric indicator used worldwide in Growth Monitoring and Promotion (GMP) programs, that does not require knowledge of length/height [56, 92, 95]. It has also emerged that WAZ is one of the best anthropometric indicators for predicting near-term death [92]. To date, this combination has been evaluated mainly for the optimization of the identification of undernourished children at high risk of near-term death [92, 95]. The performance of the combination for identifying children with WHZ<-3 has been only evaluated in infants under 6 months [10, 26]. Hence, we show for the first time that in children aged 6 to 59 months, a case definition combining MUAC and WAZ substantially improves the detection of children with WHZ<-3. Our findings allow us to

advocate for the use of the case definition of MUAC<130 mm or WAZ<-2 for the diagnosis of severe wasting among children 6 to 59 months in settings where the accurate measurement of length/height is not possible. For infants less than 6 months, our findings support the use of the case definition recently proposed in the updated WHO guidelines for the prevention and management of wasting as the combination of WAZ<-2 or MUAC<110 mm had the best diagnostic performance to the other case definitions evaluated [10, 96].

Although our study focused on identifying children with WHZ<-3, it worth mentioning that the proposed case definition of MUAC<130 mm or WAZ<-3 will be very useful in identifying children concurrently severely wasted and stunted in Timor-Leste, and likely in all settings with a high proportion of stunted under-five children. This characteristic may greatly benefit treatment outcomes, as the concurrence of multiple anthropometric deficits increases the risk of near-term death [7, 8, 95]. Also growing evidence shows that WAZ is a good predictor of mortality in both infants aged less than 6 months and children aged 6 to 59 months [26, 97, 98]. Fortunately, it has been demonstrated that underweight children and stunted children who are also wasted do recover when treated with the current IMAM protocol, although those with low WAZ recover more slowly [99, 100]. Moreover, adopting this case definition may strengthen the linkage between the IMAM and GMP programs, thereby improving the integration of preventative and curative interventions to address undernutrition, including wasting and stunting.

This study was conducted in a context where edematous malnutrition is rare; hence, the optimal cut-offs observed may not be appropriate in zones where there is a high proportion of the mixed form of SAM called kwashiorkor-marasmic such as in central Africa [101]. It was recently shown that children with combined SAM (WHZ<-3 and bilateral pitting edema) tend to have a lower MUAC measurement than those with severe wasting alone, suggesting that MUAC average of children with WHZ<-3 is likely to be lower in these countries than in countries with a lower proportion of mixed forms of SAM [101]. However, it is unlikely that the clinical characteristics of children with the mixed form of SAM would significantly affect the identification of children with WHZ<-3 and their enrolment in treatment programs when using the proposed cut-offs and case definitions, as edema is already an independent criterion for initiating treatment.

Our results show that children with WHZ<-3 not identified by the case definition MUAC<130 mm or WAZ<-3 were all not stunted suggesting they were the tallest of the group and that the proposed case definition may be less accurate in children of the ectomorphic body shape phenotype who are typically lean and tall and for whom there is a controversy on the significance and associated risk of the low WHZ [19, 41, 102]. Based on the MUAC cut-offs associated with increased near-term death in different countries, children with a MUAC ≥130 mm likely have a very low risk of near-term death [93, 103]. However, this hypothesis needs to be verified to confirm the safety of this case definition in all contexts.

The main strength of our study is that we analyzed data from a country-representative national nutrition survey. The second strength is the large sample size and the availability of all the variables needed for analyses, with less than 2% of the entries excluded for missingness or improbable values. These strengths allow to confidently use our findings for suggesting changes in national policies. The main limitation of our study is the cross-sectional nature of the original survey, meaning that our conclusions are not based on the risk of mortality which is the criterion used for the selection of the current WHO recommended MUAC cut-offs and case definitions. Nevertheless, as our primary objective was to identify a case definition that maximizes the identification and enrolment of children with WHZ<-3, this limitation does not affect the validity of our conclusions.

## Conclusions and programmatic implications

This study confirms the poor overlap of MUAC and WHZ in under-five Timorese children, and leads us to the conclusion that it is challenging to achieve a high level of diagnostic accuracy for identifying children with WHZ<-3 Z-score solely through expanding the MUAC cut-off. In settings where it is challenging to have accurate length/height measurements, a combination of MUAC and WAZ indicators is necessary to maximize the identification of children with WGZ<-3 Z-score. Our findings allow us to advocate for the adoption and integration into IMAM protocols the following case definitions as criteria for screening for severe wasting and admission for therapeutic feeding: MUAC<110 mm or WAZ <-2 Z-Score for infants below 6 months of age, and MUAC<130 mm or WAZ<-3 Z-Score for children aged between 6 and 59 months. Further research is needed to confirm the suitability of these case definitions in programmatic context and understand the short- and long-term outcomes of severely wasted children not being detected by the proposed case definitions. Efforts to determine a global MUAC cut-off for use in contexts that use the MUAC as the sole anthropometric criterion, should continue. These efforts should aim to increase the geographical scope of the evidence, and enable the conduct of a methodologically robust meta-analysis that could produce a cut-off value around which a global consensus can be reached.

## Supporting information

**S1 Fig. Distribution of the sex and age groups in the different municipalities.**
(TIF)

**S1 Data. Raw data used for producing the results presented in this paper.**
(DTA)

## Acknowledgments

The authors would like to thank Grace Heymiesfeld, Jose Luis Alvarez Moran, Paul Binns, and Roland Kupka for their insightful comments that improved the manuscript. We also thank the Timor-Leste Ministry of Health for granting permission to use the survey data. Thanks also to the UNICEF Timor-Leste, UNICEF East Asia and the Pacific Regional Office (EAPRO), and Action Against Hunger teams for the administrative and technical support provided throughout the study implementation.

## Author Contributions

**Conceptualization:** Mueni Mutunga, Faraja Chiwile, Natalia dos Reis de Araujo Moniz, Paluku Bahwere.

**Data curation:** Faraja Chiwile, Natalia dos Reis de Araujo Moniz, Paluku Bahwere.

**Formal analysis:** Paluku Bahwere.

**Funding acquisition:** Mueni Mutunga, Faraja Chiwile.

**Methodology:** Paluku Bahwere.

**Project administration:** Mueni Mutunga, Faraja Chiwile.

**Writing – original draft:** Paluku Bahwere.

**Writing – review & editing:** Mueni Mutunga, Faraja Chiwile, Natalia dos Reis de Araujo Moniz.

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
