## [Decision Letter · Decision Letter 0]

21 Feb 2024

PONE-D-24-05436Improving case-detection of severe wasting among under-five-year-old children in Timor Leste: A secondary analysis of data from the 2020 national cross-sectional food and nutrition surveyPLOS ONE

Dear Dr. Bahwere,

Thank you for submitting your manuscript to PLOS ONE. After careful consideration, we feel that it has merit but does not fully meet PLOS ONE’s publication criteria as it currently stands. Therefore, we invite you to submit a revised version of the manuscript that addresses the points raised during the review process.

We look forward to receiving your revised manuscript.

Kind regards,

Guy Franck Biaou ALE, PhD

Academic Editor

PLOS ONE

Journal Requirements:

2. Please note that your Data Availability Statement is currently missing [the repository name and/or the DOI/accession number of each dataset OR a direct link to access each database]. If your manuscript is accepted for publication, you will be asked to provide these details on a very short timeline. We therefore suggest that you provide this information now, though we will not hold up the peer review process if you are unable.

3. PLOS requires an ORCID iD for the corresponding author in Editorial Manager on papers submitted after December 6th, 2016. Please ensure that you have an ORCID iD and that it is validated in Editorial Manager. To do this, go to ‘Update my Information’ (in the upper left-hand corner of the main menu), and click on the Fetch/Validate link next to the ORCID field. This will take you to the ORCID site and allow you to create a new iD or authenticate a pre-existing iD in Editorial Manager. Please see the following video for instructions on linking an ORCID iD to your Editorial Manager account: https://www.youtube.com/watch?v=_xcclfuvtxQ.

Reviewers' comments:

Reviewer's Responses to Questions

**Comments to the Author**

1. Is the manuscript technically sound, and do the data support the conclusions?

Reviewer #1: Yes

Reviewer #2: Yes

2. Has the statistical analysis been performed appropriately and rigorously? 

Reviewer #1: Yes

Reviewer #2: Yes

3. Have the authors made all data underlying the findings in their manuscript fully available?

Reviewer #1: Yes

Reviewer #2: Yes

4. Is the manuscript presented in an intelligible fashion and written in standard English?

Reviewer #1: Yes

Reviewer #2: Yes

5. Review Comments to the Author

Reviewer #1: Dear author,

This work is very fine and please discuss in detail with similar study conducted so far to show the trend . Please update old references. Please check once the grammatical and any technical error in the manuscript.

Reviewer #2: Introduction

Lines 56, 57, and 66 need references.

In line 72 write the word countries instead of counties.

In line 77, what does IMAM mean? Abbreviate it. The author abbreviates it at line 120. Kindly abbreviate it at the start.

Sometimes authors capitalize the word ‘Malnutrition’, and sometimes write it in small letters like ‘malnutrition’. Check lines 83 and 85. Use one format.

The authors used around 47 references older than 5 years. As it's well-known that a research paper should use the last 5 years of literature, to obtain the most relevant information. The percentages of the last 5 years of references were 49.5 % (46 references) while old references were 50.5% (47 references). Kindly use updated references.

Discussions

In line 353 the sentence does not end with full stop. Check punctuation marks in the whole manuscript.

In line 362, there are 2 full stops at the end of the sentence. Please correct it.

Also, check grammar mistakes.

Conclusion

Please add further directions (e.g., further studies on what should be considered).

Add one simple paragraph for easy understanding of the common person.

Please check the whole manuscript carefully before resubmitting.

6. PLOS authors have the option to publish the peer review history of their article (what does this mean?). If published, this will include your full peer review and any attached files.

Reviewer #1: **Yes: **Habtamu Fekadu Gemede

Reviewer #2: No

---

## [Author Response · Author response to Decision Letter 0]

26 Apr 2024

Dear Editors,

We have addressed all the comments in the response to reviews file that it attached to this submission.

---

## [Decision Letter · Decision Letter 1]

19 Jul 2024

Improving case-detection of severe wasting among under-five-year-old children in Timor Leste: A secondary analysis of data from the 2020 national cross-sectional food and nutrition survey

PONE-D-24-05436R1

Dear Dr. Bahwere,

We’re pleased to inform you that your manuscript has been judged scientifically suitable for publication and will be formally accepted for publication once it meets all outstanding technical requirements.

Kind regards,

Sajid Bashir Soofi

Academic Editor

PLOS ONE

Additional Editor Comments (optional):

Reviewers' comments:

Reviewer's Responses to Questions

**Comments to the Author**

1. If the authors have adequately addressed your comments raised in a previous round of review and you feel that this manuscript is now acceptable for publication, you may indicate that here to bypass the “Comments to the Author” section, enter your conflict of interest statement in the “Confidential to Editor” section, and submit your "Accept" recommendation.

Reviewer #1: All comments have been addressed

Reviewer #2: All comments have been addressed

2. Is the manuscript technically sound, and do the data support the conclusions?

Reviewer #1: Yes

Reviewer #2: Yes

3. Has the statistical analysis been performed appropriately and rigorously? 

Reviewer #1: Yes

Reviewer #2: Yes

4. Have the authors made all data underlying the findings in their manuscript fully available?

Reviewer #1: Yes

Reviewer #2: Yes

5. Is the manuscript presented in an intelligible fashion and written in standard English?

Reviewer #1: Yes

Reviewer #2: Yes

6. Review Comments to the Author

Reviewer #1: Dear Author,

Thank for considering my comments and suggested for publication. Please keep up the nice work and thank very much!

Reviewer #2: The abstract has been revised accordingly. References have been updated with latest ones. Introduction is also revised as suggested. conclusion is also up to the mark.

7. PLOS authors have the option to publish the peer review history of their article (what does this mean?). If published, this will include your full peer review and any attached files.

Reviewer #1: No

Reviewer #2: **Yes: **Dr. Aftab Ahmed

Associate Professor

Department of Nutritional Sciences

Government College University Faisalabad, Punjab, Pakistan

---

## [Editor Report · Acceptance letter]

12 Aug 2024

PONE-D-24-05436R1 

PLOS ONE

Dear Dr. Bahwere, 

I'm pleased to inform you that your manuscript has been deemed suitable for publication in PLOS ONE. Congratulations! Your manuscript is now being handed over to our production team.

Kind regards, 

on behalf of

Professor Sajid Bashir Soofi 

Academic Editor

PLOS ONE